# Leveraging Theoretical Tradeoffs in Hyperparameter Selection for Improved Empirical Performance

**Parikshit Ram**                                    P.RAM@ACM.ORG
**Alexander G. Gray**                    ALEXANDER.GRAY@IBM.COM
**Horst C. Samulowitz**                     SAMULOWITZ@US.IBM.COM
*IBM Research, Yorktown-Heights, USA*

## Abstract

The tradeoffs in the excess risk incurred from data-driven learning of a single model has been studied by decomposing the excess risk into approximation, estimation and optimization errors. In this paper, we focus on the excess risk incurred in data-driven hyperparameter optimization (HPO) and its interaction with approximate empirical risk minimization (ERM) necessitated by large data. We present novel bounds for the excess risk in various common scenarios in HPO. Based on these results, we propose practical heuristics that allow us to improve performance or reduce computational overhead of data-driven HPO, demonstrating over $2\times$ speedup with no loss in predictive performance in our preliminary results.

## 1. Excess Risk in Learning

The learning process has various sources of errors. Given data, we choose a model or function class $\mathcal{F}$ which corresponds to not just a method (such as Support Vector Machines, Generalized Linear Models, Neural Networks, Decision trees) but its specific *hyperparameters* (HPs) – these HPs refer to anything that would influence the predictive performance of the model learned from data. Given choice of $\mathcal{F}$, learning searches for the function via (possibly approximate) empirical risk minimization (ERM). We currently have an understanding of the factors affecting the *excess risk* of this chosen function – (i) the choice of the function class and its capacity to model the process generating the data, (ii) the use of empirical error instead of the generalization error, and (iii) the approximation in the minimization of the empirical error (Vapnik, 2006; Devroye et al., 2013; Bottou and Bousquet, 2008).

However, in practice, a significant part of the whole exercise is the choice of the function class $\mathcal{F}$ (method and its HPs). Usually, we consider a (possibly large) set of function classes and select one of them based on the data-driven *hyperparameter optimization* (HPO) or *model selection*. This search can be done via grid search. AutoML (automated machine learning) has spurred a lot of research in efficient HPO (Hutter et al., 2011; Shahriari et al., 2016). The automation allows us to efficiently explore larger numbers of HP configurations for improved performance. HPO has been extended from model configurations to the configuration of complete ML pipelines with efficient Combined Algorithm Selection and HPO (or CASH) algorithms (Thornton et al., 2012; Feurer et al., 2015; Kotthoff et al., 2017; Rakotoarison et al., 2019; Liu et al., 2020; Sarigiannis et al., 2019).

Penalty-based model selection has been thoroughly studied theoretically, resulting in strong guarantees in the form of "oracle inequalities" – the expected excess risk of the selected model (or HP) is within a multiplicative and additive factor of the best possible

excess risk if an oracle provided us with the best HP. This has been widely studied in (binary) classification (Boucheron et al., 2005), (bounded) regression and density estimation (Massart, 2007; Arlot et al., 2010). However, in practice, penalty-based model selection is not used for data-driven HPO, and we resort to cross-validation (CV) because they are more universally applicable. CV schemes have been shown to be theoretically competitive to penalty-based selection at the cost of having less data for learning. We focus on CV widely used in the aforementioned HPO. While CV based model selection has been studied theoretically, there are various scenarios in practical HPO, which have not been explored in literature. Here, we focus on such a scenario – the practice of learning a final model with the HP selected via CV on the full data, and the role of approximate ERM in HPO and the final model training:

▶ We study the HPO process under the different scenarios and identify new sources of excess risk introduced by this process, and provide upper bounds to the excess risk.

▶ We propose data-driven practical heuristics based on these bounds to improve the predictive performance of selected model and reduce the computational overhead when performing HPO with approximate ERM.

**Outline.** In §2, we discuss existing excess risk decompositions and describe our HPO setting. We study the excess risk resulting from HPO in §3, providing novel bounds, and deriving practical heuristics from these bounds. We conclude with future work in §4.

## 2. Existing Decompositions & Model Selection

For a particular model, let $\mathcal{F}_\lambda$ denote the function class for some HP $\lambda \in \Lambda$ in the space of all possible valid HPs $\Lambda$. For any function $f : \mathcal{X} \to \mathcal{Y}$ with $(X, Y), X \subset \mathcal{X}, Y \subset \mathcal{Y}$ generated from a distribution $P$, and a $B$-bounded loss function $\ell : \mathcal{Y} \times \mathcal{Y}$, the expected risk $E(f)$ and the empirical risk $E_n(f)$ with $n$ samples $\{(x_i, y_i)\}_{i=1}^n \sim P^n$ of the model $f$ is given by

$$E(f) = \int \ell(y, f(x)) dP(x, y), \quad E_n(f) = \frac{1}{n} \sum_{i=1}^n \ell(y_i, f(x_i)). \tag{1}$$

We denote the Bayes optimal prediction function as $f^\star$ such that, for any $(x, y) \sim P$, $f^\star(x) = \arg\min_{\hat{y}} \mathbb{E}[\ell(y, \hat{y})|x]$. Let $\bar{f}_\lambda = \arg\min_{f \in \mathcal{F}_\lambda} E(f)$ be the true risk minimizer in $\mathcal{F}_\lambda$, while $\hat{f}_{n,\lambda} = \arg\min_{f \in \mathcal{F}_\lambda} E_n(f)$ be the ERM solution. With ERM over $\mathcal{F}_\lambda$, the excess risk incurred $\mathcal{E} = E(\hat{f}_{n,\lambda}) - E(f^\star)$ decomposes into two terms – (i) the *approximation error* $\mathcal{E}_{app}(\lambda) = E(\bar{f}_\lambda) - E(f^\star)$, and (ii) the *estimation error* $\mathcal{E}_{est}(\lambda) = E(\hat{f}_{n,\lambda}) - E(\bar{f})$. For limited number of samples $n$, there is a tradeoff between $\mathcal{E}_{app}$ and $\mathcal{E}_{est}$, where a larger function class $\mathcal{F}_\lambda$ usually reduces $\mathcal{E}_{app}$ but increases $\mathcal{E}_{est}$ (Vapnik, 2006; Devroye et al., 2013). See §A (and Table 1) for notations, precise definitions and exact existing decomposition. Bottou and Bousquet (2008) study the tradeoffs in a "large-scale" setting where the learning has a computational budget and they categorize the learning setting into two following scales: (i) "small-scale": the number of samples $n$ is small enough to allow for exact ERM within the compute budget, (ii) "large-scale": the ERM needs to be approximated given the compute budget. They study the excess risk of an approximate ERM solution $\widetilde{f}_{n,\lambda} \in \mathcal{F}_\lambda$, introducing the *optimization error* $\mathcal{E}_{opt}(\lambda) = E(\widetilde{f}_{n,\lambda}) - E(\hat{f}_{n,\lambda})$ – the excess risk incurred due to approximate ERM – and argue that, in large-scale learning, approximate ERM on all $n$ samples has better excess risk than exact ERM on a subsample of size $n' \le n$.

We wish to understand the excess risk in HPO over $\Lambda$. Limited to $n$ samples from the data distribution $P$, the HPO problem with CV is given by this bilevel optimization:

$$\textit{Outer: } \hat{\lambda} = \arg\min_{\lambda \in \Lambda} E_{n_v}^v(\hat{g}_{n_t,\lambda}), \quad \textit{Inner: } \hat{g}_{n_t,\lambda} = \arg\min_{f \in \mathcal{F}_\lambda} E_{n_t}(f). \tag{2}$$

This results in the selection of the HP $\hat{\lambda}$. The inner level problem for a fixed HP $\lambda$ is solved via ERM with $E_{n_t}(\cdot)$ on $n_t \leq n$ samples, while the outer level problem considers an objective $E_{n_v}^v(\cdot)$ which is evaluated using $n_v \leq n$ samples not used for the inner ERM – while $E_{n_t}(\cdot)$ and $E_{n_v}^v(\cdot)$ might have the same form, the $\cdot^v$ superscript highlights their difference.

This formulation incorporates the common practice of splitting the samples into a *training* set and a *held-out validation* set (of sizes $n_t, n_v \leq n$ with usually $n_t + n_v = n$) and the best HP is selected based on empirical risk of the solution of ERM (with $n_t$ samples) on the heldout set (of size $n_v$). When performing $k$-*fold cross-validation*, the inner optimization is solved $k$ times for each HP $\lambda$ (on $k$ different sets of $n_t$ samples), and the outer optimization averages the objective on $k$ held-out sets (of size $n_v$) across the $k$ learned models. In this paper, we limit our scope to a single training/validation split.

## 3. Excess Risk In HP Selection

We first analyze HPO with hold-out validation in the "small scale" setting and develop novel results for a common practice in model selection. Using these results, we propose a practical heuristic to obtain improved performance in data-driven small-scale HPO and empirically highlight its utility. We then build upon these results to provide excess risk bounds in the "large scale" setting necessitating approximate ERM. Based on these, we propose another heuristic that allows us to enjoy the predictive performance of HPO with exact ERM at reduced computational cost. We also demonstrate the practical utility of this heuristic.

In our ensuing presentation, the *maximum error discrepancy* $\sup_{f \in \mathcal{F}_\lambda} |E_n(f) - E(f)|$ of any function class $\mathcal{F}_\lambda$ will be appearing multiple times. For ease of exposition, we consider as its generic upper-bound $\Delta(\mathcal{F}_\lambda, n, \delta)$ with probability at least $1 - \delta$, $\delta > 0$. Precise definitions of $\Delta(\mathcal{F}_\lambda, n, \delta)$ can be achieved using covering numbers (Corollary 6 in §B) or Radamacher complexities (Theorem 7 in §B). This $\Delta(\mathcal{F}_\lambda, n, \delta)$ is monotonically non-increasing with $n$ and is indicative of the "size" or complexity of the class $\mathcal{F}_\lambda$ – a more complex class will have higher maximum error discrepancy.

### 3.1 Small-scale Learning

In the small-scale learning setup, after the selection of $\hat{\lambda}$ by solving problem (2), either (**Case A**) the ERM solution $\hat{g}_{n_t,\hat{\lambda}}$ on the training split in the inner problem (2) is utilized as is, or (**Case B**) ERM is performed with all the $n$ samples available on $\mathcal{F}_{\hat{\lambda}}$ to obtain the ERM solution $\hat{f}_{n,\hat{\lambda}} = \arg\min_{f \in \mathcal{F}_{\hat{\lambda}}} E_n(f)$. We study the different factors affecting the excess risk introduced in both these cases. See §B.4 for technical details and proofs.

**Case A (No model discrepancy).** This case is commonly used with deep neural networks where the models trained during HP (and architecture) selection are the ones finally utilized for deployment. In the following theorem, we provide a bound on the excess risk:

**Theorem 1** *Let* $L = 2|\Lambda| + 2$. *Then, with probability at least* $1 - \delta$ *for any* $\delta > 0$, *the excess risk* $\mathcal{E} = E(\hat{g}_{n_t,\hat{\lambda}}) - E(f^\star)$ *is bounded from above as:*

$$\mathcal{E} \leq \min_{\lambda \in \Lambda} \left\{ 2\Delta\left(\mathcal{F}_\lambda, n_t, \delta/L_1\right) + \mathcal{E}_{\mathsf{app}}(\lambda) \right\} + B\sqrt{2\log(L_1/\delta)/n_v}. \tag{3}$$

We present this simple result for completeness to highlight how we will extend this result to provide bounds for the model selection scenarios we consider in the following.

**Case B (Model discrepancy).** This case is common in ML where we wish to make the most of the available data and hence train the model for the selected HP $\hat{\lambda}$ with the full training set of $n$ samples. However, this introduces a discrepancy – the learned model $\hat{g}_{n_t,\hat{\lambda}}$ that guides the HP selection is different from the model $\hat{f}_{n,\hat{\lambda}}$ whose excess risk we wish to understand. We provide a novel bound for the excess risk incurred in this process:

**Theorem 2** *Let* $L_2 = 2|\Lambda| + 3$. *Let* $\mathcal{I}_{n,n_t,\hat{\lambda}} = E_n(\hat{g}_{n_t,\hat{\lambda}}) - E_n(\hat{f}_{n,\hat{\lambda}})$ *denote the "empirical risk improvement" obtained by refitting the model on the full training set. Then, with probability at least* $1 - \delta$ *for any* $\delta > 0$, *the excess risk* $\mathcal{E} = E(\hat{f}_{n,\hat{\lambda}}) - E(f^\star)$ *is bounded from above by:*

$$\mathcal{E} \leq \min_{\lambda \in \Lambda} \left\{ 2\Delta\left(\mathcal{F}_\lambda, n_t, \delta/L_2\right) + \mathcal{E}_{\mathsf{app}}(\lambda) \right\} - \mathcal{I}_{n,n_t,\hat{\lambda}} + B\sqrt{2\log(L_2/\delta)}\left(1/\sqrt{n} + 1/\sqrt{n_v}\right). \tag{4}$$

Note that $\mathcal{I}_{n,n_t,\hat{\lambda}} \geq 0$ by definition of $\hat{f}_{n,\hat{\lambda}}$ being the ERM solution in $\mathcal{F}_{\hat{\lambda}}$. Comparing to Case A in Theorem 1 to Theorem 2, we see the Case B involves more computation (training $\hat{f}_{n,\hat{\lambda}}$) but is only preferable over Case A if the (computable) empirical risk improvement $\mathcal{I}_{n,n_t,\hat{\lambda}}$ is relatively significant, reducing the excess risk bound by that quantity. An interesting aspect of this result is that the statistical cost of the additional ERM for $\hat{f}_{n,\hat{\lambda}}$ is $O(1/\sqrt{n})$ but does not depend on the size of $\mathcal{F}_{\hat{\lambda}}$. Based on the results, we propose a practical heuristic to select between $\hat{f}_{n,\hat{\lambda}}$ and $\hat{g}_{n_t,\hat{\lambda}}$ as the final model to be deployed for improved excess risk:

**Heuristic 1** *Let us define the following data-dependent scalars* $\alpha, \beta$ *based on the quantities in Theorems 1 & 2, and we select* $\hat{f}_{n,\hat{\lambda}}$ *as the final model if* $\alpha \geq \beta$, *or select* $\hat{g}_{n_t,\hat{\lambda}}$ *otherwise:*

$$\alpha = B\sqrt{2\log(L_1/\delta)/n_v}, \quad \beta = -\mathcal{I}_{n,n_t,\hat{\lambda}} + B\sqrt{2\log(L_2/\delta)}\left(1/\sqrt{n} + 1/\sqrt{n_v}\right). \tag{5}$$

### 3.1.1 Empirical validation

For the purposes of empirical validation of our results and proposed heuristic, we consider a HPO problem with 36 neural network configurations ($|\Lambda| = 36$) on a synthetic classification data set with Bayes optimal risk $E(f^*) = 0$. We use synthetic data to have control over the experiment and generate large fresh samples to accurately estimate true risks $E(f)$ of any model $f$. The estimation of $\mathcal{E}_{\mathsf{app}}(\lambda)$ and $\Delta(\mathcal{F}_\lambda, n, \delta)$ is detailed in §B.6. We consider sample sizes $n \in [2^9, 2^{14}]$ and different values for $\frac{n_v}{n} \in [0.1, 0.5]$ with $n_t = n - n_v$. We present a subset of the results in Figure 1 (all results in Figure 3 in §B.6). The results are averaged over 10 trials. We set the failure probability $\delta = 0.05$.

**Result (i).** We compare the excess risk of $\hat{g}_{n_t,\hat{\lambda}}$ and $\hat{f}_{n,\hat{\lambda}}$ (solid green & blue respectively) and their bounds from Theorems 1 and 2 (dashed green & blue respectively) in Figure 1a for varying values of $n$ for 2 values of $\frac{n_v}{n}$. In one case (Figure 1a left), the excess risk of $\hat{g}_{n_t,\hat{\lambda}}$

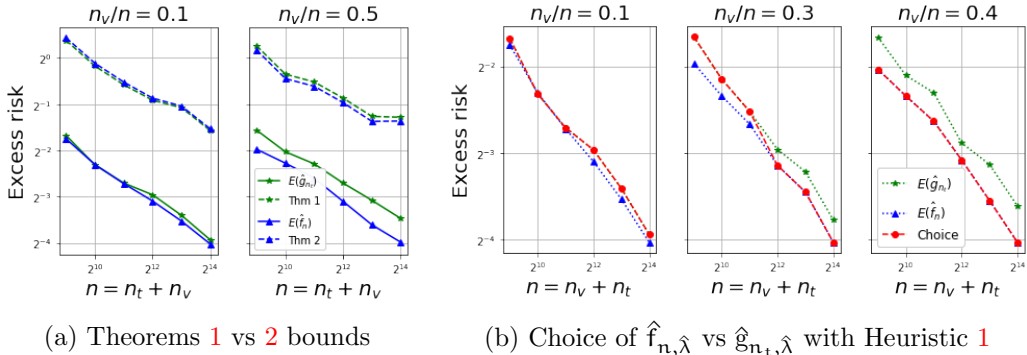

(a) Theorems 1 vs 2 bounds

(b) Choice of $\hat{f}_{n,\hat{\lambda}}$ vs $\hat{g}_{n_t,\hat{\lambda}}$ with Heuristic 1

Figure 1: Empirical validation for Theorem 1 and Theorem 2 and proposed Heuristic 1.

and $\hat{f}_{n,\hat{\lambda}}$ are close, and so are their respective bounds. In another case (Figure 1a right), $\hat{f}_{n,\hat{\lambda}}$ has better excess risk than $\hat{g}_{n_t,\hat{\lambda}}$, and that behavior is reflected in their respective bounds, with the Theorem 2 bound being slightly lower. This result indicates that the common practice of generating $\hat{f}_{n,\hat{\lambda}}$ *can* sometimes lead to tighter excess risk bounds.

**Result (ii).** We explore the practical utility of Heuristic 1. Figure 1b compares this heuristic "Choice" (dashed red) to $\hat{f}_{n,\hat{\lambda}}$ & $\hat{g}_{n_t,\hat{\lambda}}$ (dotted blue & green) – in most cases (different values of $n$, $\frac{n_v}{n}$), Heuristic 1 makes the best choice between $\hat{f}_{n,\hat{\lambda}}$ and $\hat{g}_{n_t,\hat{\lambda}}$.

### 3.2 Large-scale Learning

We now extend our analysis to the "large-scale" setting. In HPO, all the ERMs are solved approximately – the ERM for the inner problems in (2) during the selection of $\hat{\lambda}$ is performed to $\rho_{\texttt{in}}$ tolerance to get approximate ERM solutions $\widetilde{g}_{n_t,\lambda}, \lambda \in \Lambda$, where $E_{n_t}(\widetilde{g}_{n_t,\lambda}) - E_{n_t}(\hat{g}_{n_t,\lambda}) \leq \rho_{\texttt{in}}$; once $\hat{\lambda}$ is selected, the final ERM over $\mathcal{F}_{\hat{\lambda}}$ with all $n$ samples is performed to $\rho_{\texttt{out}}$ tolerance to get $\widetilde{f}_{n,\hat{\lambda}}$ with $E_n(\widetilde{f}_{n,\hat{\lambda}}) - E_n(\hat{f}_{n,\hat{\lambda}}) \leq \rho_{\texttt{out}}$. Then, in the large scale setting, the HPO becomes the following problem:

$$\hat{\lambda} = \arg\min_{\lambda \in \Lambda} E_{n_v}^v(\widetilde{g}_{n_t,\lambda}), \quad \widetilde{g}_{n_t,\lambda} \in \left\{ g \in \mathcal{F}_\lambda \colon E_{n_t}(g) \leq \min_{f \in \mathcal{F}_\lambda} E_{n_t}(f) + \rho_{\texttt{in}} \right\} \qquad (6)$$

Similar to the "small-scale learning" setting, we can bound the excess risk with and without model discrepancy. Here we present the no-model-discrepancy case, with the model-discrepancy case in Theorem 10 (and proofs) in §B.5 for lack of space:

**Theorem 3** *The excess risk $\mathcal{E} = E(\widetilde{g}_{n_t,\hat{\lambda}}) - E(f^*)$ can be bounded from above with probability at least $1 - \delta$ for any $\delta > 0$ and $L_3 = (2 + 3|\Lambda|)$:*

$$\mathcal{E} \leq \min_{\lambda \in \Lambda} \left\{ 2\Delta\left(\mathcal{F}_\lambda, n_t, \delta/L_3\right) + \mathcal{E}_{\texttt{app}}(\lambda) \right\} + B\sqrt{2\log(L_3/\delta)/n_v} + \rho_{\texttt{in}}. \qquad (7)$$

With no model discrepancy (Theorem 3), the approximate ERM in the inner problem (6) introduces an additional $\rho_{\texttt{in}}$ term to the excess risk compared to Theorem 1. In the presence of model discrepancy (Theorem 10), the approximation in the ERM shows up in two different ways – $\rho_{\texttt{in}}$ from the approximation in the inner problem (6) similar to the no-model-discrepancy case, and $\rho_{\texttt{out}}$ from the approximation in the final ERM.

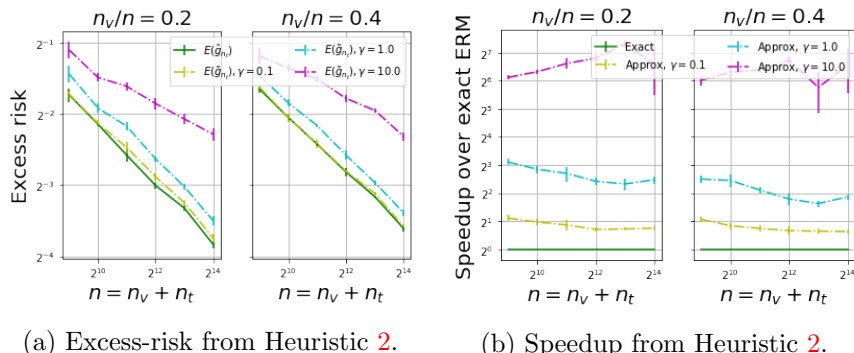

(a) Excess-risk from Heuristic 2.  (b) Speedup from Heuristic 2.

Figure 2: Empirical validation of the utility of Heuristic 2 for data-dependent choice of $\rho_{\texttt{in}}$

For any problem, $\rho_{\texttt{in}}$ and $\rho_{\texttt{out}}$ are set by the user (possibly based on available compute budget). However, in HPO with approximate ERM, *we can leverage the excess risk bounds to make a more informed choice.* Based on Theorem 3, we can see that $\rho_{\texttt{in}}$ needs to be of the same order of magnitude as the other terms in (7) – any smaller $\rho_{\texttt{in}}$ will not improve the excess risk significantly; any larger magnitude of $\rho_{\texttt{in}}$ will make this optimization error the dominant term in the excess risk. In practice, we cannot compare $\rho_{\texttt{in}}$ and $\rho_{\texttt{out}}$ to the terms $\mathcal{E}_{\texttt{app}}(\lambda)$ and $\Delta(\mathcal{F}_\lambda, n_t, \delta)$. However, we ***can*** compare them to $B\sqrt{2\log(L_3/\delta)/n_v}$ – if $\rho_{\texttt{in}} \gg B\sqrt{2\log(L_3/\delta)/n_v}$, $\rho_{\texttt{in}}$ might dominate the excess risk, and it might be beneficial to allocate more resources (if possible) to reduce $\rho_{\texttt{in}}$ to $O(1/\sqrt{n_v})$; however, reducing $\rho_{\texttt{in}}$ any further will not significantly improve the excess risk. Based on this observation, we propose another heuristic for HPO with approximate ERM with regards to the choice of $\rho_{\texttt{in}}$:

**Heuristic 2** *Based on the terms defined in Theorem 3, select a scaling parameter $\gamma > 0$ and set $\rho_{\texttt{in}}$ as $\rho_{\texttt{in}} = \gamma B\sqrt{2\log(L_3/\delta)/n_v}$ such that $\rho_{\texttt{in}} \sim o(B\sqrt{2\log(L_3/\delta)/n_v})$. A value of $\gamma = 0.1$ suffices in our experience.*

### 3.2.1 EMPIRICAL VALIDATION

To demonstrate the practical utility of the proposed Heuristic 2, we continue with the aforementioned HPO problem over 36 neural network configurations on a synthetic classification data. We consider three choices for $\rho_{\texttt{in}} = \gamma\sqrt{2\log(L_3/\delta)/n_v}$ with $\gamma \in \{0.1, 1, 10\}$. We set $\delta = 0.05$ and consider different values of $n$ and $\frac{n_v}{n}$ (see Figure 4 in §B.6 for all results).

Figure 2 compares the performance of the exact ERM HPO using $\hat{g}_{n_t,\lambda}, \lambda \in \Lambda$ to approximation ERM HPO for different $\rho_{\texttt{in}}$, using $\widetilde{g}_{n_t,\lambda}$ instead. In Figure 2a, we compare the excess risk incurred from approximate ERM with the data dependent choice of $\rho_{\texttt{in}}$ compared to exact ERM. We see that $\gamma = 0.1$ leads to a sufficiently small $\rho_{\texttt{in}}$ that matches the predictive performance of exact ERM. Any smaller approximation $\rho_{\texttt{in}}$ would not improve the excess risk. The results also indicate that $\gamma = 10$ leads to a $\rho_{\texttt{in}}$ where the optimization error dominates the excess risk, implying that $\rho_{\texttt{in}}$ should be reduced if possible. Figure 2b presents the computational speedups obtained for the corresponding data-dependent choices of $\rho_{\texttt{in}}$ – we see that we can get a $2\times$ speedup over exact ERM without any degradation in excess risk with $\gamma = 0.1$ while obtaining around $4-6\times$ speedup with slight degradation in

performance with $\gamma = 1$. These results provide empirical evidence for the practical utility of the proposed Heuristic 2 obtained from Theorem 3 – *the proposed heuristic provides a data driven way of setting the ERM approximation level in HPO.*

## 4. Conclusion

We study excess risk in different HP selection scenarios. The results allow us to make more informed decisions regarding choices in HPO. As future work, we wish to study similar tradeoffs in Bayesian optimization based HPO (Shahriari et al., 2016). We also plan to study multi-fidelity HPO schemes such as Successive Halving (Jamieson and Talwalkar, 2016) where the ERM approximation in HPO is adaptively modified.

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

## Appendix A. Detailed notation and current decompositions

Consider samples $(x, y)$ of covariates $x$ and dependent variable $y$ from a joint distribution $P(x, y)$ over $\mathcal{X} \times \mathcal{Y}$ and loss function $\ell : \mathcal{Y} \times \mathcal{Y} \rightarrow \mathbb{R}$. The conditional distribution $P(y|x)$ encodes the relationship between the covariates $x$ and the dependent variable $y$. In this setup, the Bayes optimal predictor is given by

$$f^\star : \mathcal{X} \rightarrow \mathcal{Y} \text{ such that } f^\star(x) = \arg\min_{\hat{y} \in \mathcal{Y}} \mathbb{E}\left[\ell(y, \hat{y})|x\right]. \tag{8}$$

For any model/function $f : \mathcal{X} \rightarrow \mathcal{Y}$, the expected risk of the model is given by

$$E(f) = \int \ell(y, f(x)) dP(x, y) = \mathbb{E}\left[\ell(y, f(x))\right]. \tag{9}$$

Given a model class $\mathcal{F}$, the true risk minimizer is $\bar{f} = \arg\min_{f \in \mathcal{F}} E(f)$. However, this requires access to the distribution $P(x, y)$. In general, we consider $n$ samples $\{(x_i, y_i)\}_{i=1}^n \sim P(x, y)$. The empirical risk is given by

$$E_n(f) = \frac{1}{n} \sum_{i=1}^n \ell(y_i, f(x_i)) = \mathbb{E}_n\left[\ell(y, f(x))\right], \tag{10}$$

and the empirical risk minimizer is $\hat{f}_n = \arg\min_{f \in \mathcal{F}} E_n(f)$.

The excess risk $\mathcal{E}$ incurred by the empirical risk minimizer $\hat{f}_\mathcal{F}$ is given by

$$\mathcal{E} = E(\hat{f}_n) - E(f^\star) \tag{11}$$

$$= \underbrace{E(\hat{f}_n) - E(\bar{f})}_{\mathcal{E}_{\text{est}}} + \underbrace{E(\bar{f}) - E(f^*)}_{\mathcal{E}_{\text{app}}}. \tag{12}$$

The *approximation error* $\mathcal{E}_{\text{app}}$ is unknown in general but is assumed to reduce as the "size" of the class $\mathcal{F}$ grows. The *estimation error* $\mathcal{E}_{\text{est}}$ is bounded from above by $2\Delta(\mathcal{F}, n, \delta)$ with probability at least $1 - 2\delta$ (see Theorem 9).

In practice, we solve the empirical risk minimization to some approximation, and we care about the excess risk incurred by an approximate empirical risk minimizer $\widetilde{f}_n$. The excess risk of $\widetilde{f}_n$ is given by

$$\mathcal{E} = E(\widetilde{f}_n) - E(f^\star) \tag{13}$$

$$= \underbrace{E(\widetilde{f}_n) - E(\hat{f}_n)}_{\mathcal{E}_{\text{opt}}} + \underbrace{E(\hat{f}_n) - E(\bar{f})}_{\mathcal{E}_{\text{est}}} + \underbrace{E(\bar{f}) - E(f^*)}_{\mathcal{E}_{\text{app}}}. \tag{14}$$

This decomposition of the excess risk includes a new *optimization error* $\mathcal{E}_{\text{opt}}$ (Bottou and Bousquet, 2008). If the approximate empirical risk minimizer satisfies the following for some $\rho > 0$ with probability at least $1 - \delta$,

$$\widetilde{f}_n \in \left\{f \in \mathcal{F} : E_n(f) \leq E_n(\hat{f}) + \rho\right\}. \tag{15}$$

Table 1: Table of symbols

| Symbol | Description |
|---|---|
| $E(f)$ | True risk of any model $f$ |
| $E_n(f)$ | Empirical risk of any model $f$ with $n$ samples |
| $\Lambda$ | Set of L HPs $\lambda$, $L = |\Lambda|$ |
| $\mathcal{F}_\lambda$ | Model class for HP $\lambda$ |
| $f^\star$ | Bayes optimal predictor |
| $\bar{f}_\lambda$ | True risk minimizer in $\mathcal{F}_\lambda$: $\arg\min_{f \in \mathcal{F}_\lambda} E(f)$ |
| $\hat{f}_{n,\lambda}$ | Empirical risk minimizer in $\mathcal{F}_\lambda$ with $n$ samples: $\arg\min_{f \in \mathcal{F}_\lambda} E_n(f)$ |
| $\widetilde{f}_{n,\lambda}$ | Approx. empirical risk minimizer in $\mathcal{F}_\lambda$ with $n$ |
| $\hat{\lambda}$ | Solution to empirical HPO |
| $\hat{g}_{n_t,\lambda}$ | Empirical risk minimizer in $\mathcal{F}_\lambda$ with $n_t$ samples: $\arg\min_{f \in \mathcal{F}_\lambda} E_{n_t}(f)$ |
| $\widetilde{g}_{n_t,\lambda}$ | Approx. empirical risk minimizer in $\mathcal{F}_\lambda$ with $n_t$ samples |

## Appendix B. Details for Section 3

### B.1 Standard results and definitions

**Definition 4** *Let $\mathcal{F}$ be a class of functions $X$ to $Y$. For any $\epsilon > 0$, the associated $\ell_\infty$ covering number $\mathcal{N}(\mathcal{F}, \epsilon)$ of $\mathcal{F}$ is the minimal possible integer $k$ such that $\mathcal{F}$ can be covered by $k$ balls of $\ell_\infty$ radius $\epsilon$.*

We have the following standard result in statistical learning theory

**Theorem 5** *Let $\mathcal{F}$ be a class of functions from $X$ to $Y$ and let $E : Y' \times Y \to [0, B]$ be an L-Lipschitz, B-bounded risk function. Then, for any distribution $P$ over $X \times Y$,*

$$\Pr_{\{(x_i, y_i)\}_{i=1}^n \sim P^n} \left( \sup_{f \in \mathcal{F}} |E_n(f) - E(f)| \geq 3\epsilon \right) \leq 2\mathcal{N}(\mathcal{F}, \epsilon) \exp\left( -\frac{n\epsilon^2}{2B^2} \right). \tag{16}$$

**Corollary 6** *Under conditions of Definition 4 and Theorem 5, with probability at least $1 - \delta$,*

$$\sup_{f \in \mathcal{F}} |E_n(f) - E(f)| \leq \Delta(\mathcal{F}, n, \delta) = \inf_{\epsilon > 0} \left\{ 3\epsilon + B \sqrt{\frac{2}{n} \left( \log \mathcal{N}(\mathcal{F}, \epsilon) + \log(1/\delta) \right)} \right\}. \tag{17}$$

**Theorem 7** *(Bartlett and Mendelson (2002)) Let $P$ be a distribution over $X \times Y$ and let $\ell : Y' \times Y$ (where $Y \subseteq Y' \subset \mathbb{R}$) be a B-bounded loss function that is L-Lispschitz in its first argument. Let $\mathcal{F}$ be a class of functions from $X \to Y$. Then for any $\delta > 0$, with probability at least $1 - \delta$ (over the random sample draw from $P$), the following is true:*

$$\sup_{f \in F} |E_n(f) - E(f)| \leq \Delta(\mathcal{F}, n, \delta) = 4 \cdot L \cdot \mathcal{R}_n(\mathcal{F}) + 2 \cdot B \cdot \sqrt{\frac{\log(1/\delta)}{2n}}, \tag{18}$$

*where $\mathcal{R}_n(\mathcal{F})$ is the Radamacher complexity of the function class $\mathcal{F}$.*

## B.2 Notation

To concisely and conveniently prove results with somewhat similar arguments, we define the following notation for providing upper bounds:

**Definition 8** *For any quantity* $A$ *and* $B$ *such that* $A - B \bowtie \epsilon$ *with probability at least* $1 - \delta$ *for some numerical relational operator* $\bowtie$ *and* $\delta > 0$*, let us define the following notation:*

$$A \xrightarrow[\delta]{\bowtie \epsilon} B \tag{19}$$

For example, based on Corollary 6, for any function $f$ in some class $\mathcal{F}$ and $m$ samples, we can write:

$$E(f) \xrightarrow[\delta]{\leq \Delta(\mathcal{F}, m, \delta)} E_m(f) \tag{20}$$

where $\leq$ serves at the numerical relational operator $\bowtie$, $A = E(f), B = E_m(f), \epsilon = \Delta(\mathcal{F}, m, \delta)$.

## B.3 Bounds for existing decompositions

**Theorem 9** *For any function class* $\mathcal{F}$*, the estimation error* $\mathcal{E}_{\text{est}} = E(\hat{f}_n) - E(\bar{f})$ *from the ERM solution with* $n$ *samples is bounded from above by* $2\Delta(\mathcal{F}, n, \delta)$ *with probability at least* $1 - 2\delta$*.*

**Proof** We have the following relationship:

$$E(\hat{f}_n) \xrightarrow[\delta]{\leq \Delta(\mathcal{F}, n, \delta)} E_n(\hat{f}_n) \xrightarrow{\leq 0} E_n(\bar{f}) \xrightarrow[\delta]{\leq \Delta(\mathcal{F}, n, \delta)} E(\bar{f}) \tag{21}$$

∎

## B.4 Proofs for small-scale learning setup

### B.4.1 PROOF FOR THEOREM 1

**Proof** First, consider the $\mathcal{E}_{\text{hpo}} \doteq E(\hat{g}_{n_t, \hat{\lambda}}) - E(\hat{g}_{n_t, \bar{\lambda}})$ term in $\mathcal{E} = E(\hat{g}_{n_t, \hat{\lambda}}) - E(f^\star)$. By definition of $\mathcal{F}_{\hat{\lambda}}$ and $\hat{g}_{n_t, \hat{\lambda}}$, we have the following relationship for any $\mathcal{F}_{\lambda}, \lambda \in \Lambda$ with some $\delta' > 0$:

$$E(\hat{g}_{n_t, \hat{\lambda}}) \xrightarrow[\delta']{\leq B\sqrt{\log(1/\delta')/2n_\nu}} E_{n_\nu}^\nu(\hat{g}_{n_t, \hat{\lambda}}) \xrightarrow{\leq 0} E_{n_\nu}^\nu(\hat{g}_{n_t, \lambda}) \xrightarrow[\delta']{\leq B\sqrt{\log(1/\delta')/2n_\nu}} E(\hat{g}_{n_t, \lambda}). \tag{22}$$

Now for any $\mathcal{F}_{\lambda}$ with corresponding $\lambda$, for some failure probability $\delta' > 0$, we have the following relationship:

$$E(\hat{g}_{n_t, \lambda}) \xrightarrow[\delta']{\leq \Delta(\mathcal{F}_{\lambda}, n_t, \delta')} E_{n_t}(\hat{g}_{n_t, \lambda}) \xrightarrow{\leq 0} E_{n_t}(\bar{f}_{\lambda}) \xrightarrow[\delta']{\leq \Delta(\mathcal{F}_{\lambda}, n_t, \delta')} E(\bar{f}_{\lambda}) \xrightarrow{= \mathcal{E}_{\text{app}}(\lambda)} E(f^*) \tag{23}$$

Thus, we have $\mathcal{E} \leq \mathcal{E}_{\text{hpo}} + 2\Delta(\mathcal{F}_{\lambda}, n_t, \delta) + \mathcal{E}_{\text{app}}(\lambda)$ for all $\lambda \in \Lambda$ with a failure probability at most $(2 + 2L)\delta'$ using the union bound. Minimizing the bound over $\lambda$ and setting $\delta' = \frac{\delta}{2(1+L)} = \frac{\delta}{L_1}$ gives us the result in (3). ∎

### B.4.2 PROOF FOR THEOREM 2

**Proof** For the selected function class $\mathcal{F}_{\hat{\lambda}}$, we have the following relationship given definition of $E(\cdot)$, $E_n(\cdot)$ and $\mathcal{I}_{n,n_t,\hat{\lambda}}$ using Hoeffding's inequality for some $\delta' > 0$:

Let $\mu : (X \times Y)^n \to \mathbb{R}$ be the following function:

$$\mu((x_1, y_1), \ldots, (x_n, y_n)) = E_n(\hat{f}_{n,\hat{\lambda}}) - E_n(\hat{g}_{n_t,\hat{\lambda}}), \tag{24}$$

then we can show that the function $\mu$ satisfies the bounded differences condition, that is, there existing constants $c_i, i \in [n]$ such that:

$$\sup_{\{(x_j,y_j)\}_{j \in [n]}, (x_i', y_i')} \left| \mu((x_1, y_1), \ldots, (x_i, y_i), \ldots, (x_n, y_n)) - \mu((x_1, y_1), \ldots, (x_i', y_i'), \ldots, (x_n, y_n)) \right| \le c_i. \tag{25}$$

The bounded difference inequality tells us

$$\mathbb{E}\mu((x_1, y_1), \ldots, (x_n, y_n)) \xrightarrow[\delta']{\sqrt{\frac{C \log(1/\delta)}{2}}} \mu((x_1, y_1), \ldots, (x_n, y_n)), \tag{26}$$

where $C = \sum_{i=1}^{n} c_i^2$. Given the definition of $\mu(\cdot), E_n(\cdot)$, we can show that $c_i \le 2B/n$ and hence $C = 4B^2/n$, giving us

$$\mathcal{E}_{\text{md}} = E(\hat{f}_{n,\hat{\lambda}}) - E(\hat{g}_{n_t,\hat{\lambda}}) \xrightarrow[\delta']{\le 2B\sqrt{\log(1/\delta')/2n}} E_n(\hat{f}_{n,\hat{\lambda}}) - E_n(\hat{g}_{n_t,\hat{\lambda}}) = -\mathcal{I}_{n,n_t,\hat{\lambda}} \tag{27}$$

The above combined with equations (22) & (23) gives us the following for all $\lambda \in \Lambda$ with a failure probability of at most $(3 + 2L)\delta'$ for some $\delta' > 0$:

$$\mathcal{E} \le 2B\sqrt{\frac{\log(1/\delta')}{2n_v}} + 2B\sqrt{\frac{\log(1/\delta')}{2n}} - \mathcal{I}_{n,n_t,\hat{\lambda}} + 2\Delta(\mathcal{F}_\lambda, n_t, \delta') + \mathcal{E}_{\text{app}}(\lambda). \tag{28}$$

Setting $\delta' = \delta/(3 + 2L) = \delta/L_2$ and minimizing over $\lambda \in \Lambda$ gives us the result in (4). ∎

### B.5 Proofs for large scale learning setup

### B.5.1 PROOF FOR THEOREM 3

**Proof** By definition of $\mathcal{F}_{\hat{\lambda}}$, for any $\lambda \in \Lambda$ and $\delta' > 0$, we have the following relationships:

$$E(\widetilde{g}_{n_t,\hat{\lambda}}) \xrightarrow[\delta']{\le B\sqrt{\log(1/\delta')/2n_v}} E_{n_v}^v(\widetilde{g}_{n_t,\hat{\lambda}}) \xrightarrow{\le 0} E_{n_v}^v(\widetilde{g}_{n_t,\lambda}) \xrightarrow[\delta']{\le B\sqrt{\log(1/\delta')/2n_v}} E(\widetilde{g}_{n_t,\lambda}) \tag{29}$$

For any $\mathcal{F}_\lambda, \lambda \in \Lambda$, we have the following relationships by definition of $\Delta(\cdot, \cdot, \cdot)$ and (6):

$$E(\widetilde{g}_{n_t,\lambda}) \xrightarrow[\delta']{\le \Delta(\mathcal{F}_\lambda, n_t, \delta')} E_{n_t}(\widetilde{g}_{n_t,\lambda}) \xrightarrow[\delta']{\le \rho_{\text{in}}} E_{n_t}(\hat{g}_{n_t,\lambda}) \xrightarrow{\le 0} E_{n_t}(\bar{f}_\lambda) \xrightarrow[\delta']{\le \Delta(\mathcal{F}_\lambda, n_t, \delta')} E(\bar{f}_\lambda) \xrightarrow{= \mathcal{E}_{\text{app}}(\lambda)} E(f^*) \tag{30}$$

Since the above holds for any $\lambda \in \Lambda$, putting (29) and (30) together using the union bound with $\delta' = \delta/(2 + 3|\Lambda|) = \delta/L_3$ and minimizing over $\lambda \in \Lambda$ gives us the desired result in (7) with a failure probability of at most $\delta$. ∎

### B.5.2 Excess risk for model discrepancy case with approximate ERM

**Theorem 10** *Let $\mathcal{I}_{n,n_t,\hat{\lambda}} = E_n(\hat{g}_{n_t,\hat{\lambda}}) - E_n(\hat{f}_{n,\hat{\lambda}})$ be the "empirical risk improvement" as in Theorem 2, and let $\widetilde{\mathcal{I}}_{n,n_t,\hat{\lambda}} = E_n(\widetilde{g}_{n_t,\hat{\lambda}}) - E_n(\widetilde{f}_{n,\hat{\lambda}})$ be corresponding the "approximate empirical risk improvement". The excess risk $\mathcal{E} = E(\widetilde{f}_{n,\hat{\lambda}}) - E(f^*)$ can be bounded as with probability at least $1 - \delta$ for any $\delta > 0$ and $L_4 = (7 + 3|\Lambda|)$:*

$$
\begin{aligned}
\mathcal{E} \leq \min_{\lambda \in \Lambda} &\left\{ 2\Delta\left(\mathcal{F}_\lambda, n_t, \frac{\delta}{L_4}\right) + \mathcal{E}_{\mathsf{app}}(\lambda) \right\} + \rho_{\mathsf{in}} + B'\left(\frac{1}{\sqrt{2n}} + \frac{1}{\sqrt{2n_v}}\right) \\
&- \max\left\{ \widetilde{\mathcal{I}}_{n,n_t,\hat{\lambda}}, \quad \left(\mathcal{I}_{n,n_t,\hat{\lambda}} - \rho_{\mathsf{out}} - B'\left(\frac{1}{\sqrt{2n}} + \frac{1}{\sqrt{2n_t}}\right)\right)\right\}
\end{aligned}
\tag{31}
$$

*where $B' = 2B\sqrt{\log\frac{L_4}{\delta}}$.*

**Proof** We will begin by bounding the term $E(\widetilde{f}_{n,\hat{\lambda}}) - E(\widetilde{g}_{n_t,\hat{\lambda}})$ using Hoeffding's inequality with failure probability $\delta' > 0$:

$$
E(\widetilde{f}_{n,\hat{\lambda}}) - E(\widetilde{g}_{n_t,\hat{\lambda}}) \xrightarrow[\delta']{\leq 2B\sqrt{\log(1/\delta')/2n}} E_n(\widetilde{f}_{n,\hat{\lambda}}) - E_n(\widetilde{g}_{n_t,\hat{\lambda}}) = -\widetilde{\mathcal{I}}_{n,n_t,\hat{\lambda}}
\tag{32}
$$

Also

$$
E(\widetilde{f}_{n,\hat{\lambda}}) - E(\widetilde{g}_{n_t,\hat{\lambda}}) = \underbrace{E(\widetilde{f}_{n,\hat{\lambda}}) - E(\hat{f}_{n,\hat{\lambda}})}_{(A)} + \underbrace{E(\hat{f}_{n,\hat{\lambda}}) - E(\hat{g}_{n_t,\hat{\lambda}})}_{(B)} + \underbrace{E(g_{n_t,\hat{\lambda}}) - E(\widetilde{g}_{n_t,\hat{\lambda}})}_{(C)}
\tag{33}
$$

$$
(A) = E(\widetilde{f}_{n,\hat{\lambda}}) - E(\hat{f}_{n,\hat{\lambda}}) \xrightarrow[\delta']{\leq 2B\sqrt{\log(1/\delta')/2n}} E_n(\widetilde{f}_{n,\hat{\lambda}}) - E_n(\hat{f}_{n,\hat{\lambda}}) \xrightarrow[\delta']{\leq 0} \rho_{\mathsf{out}}
\tag{34}
$$

$$
(B) = E(\hat{f}_{n,\hat{\lambda}}) - E(\hat{g}_{n_t,\hat{\lambda}}) \xrightarrow[\delta']{\leq 2B\sqrt{\log(1/\delta')/2n}} E_n(\hat{f}_{n,\hat{\lambda}}) - E_n(\hat{g}_{n_t,\hat{\lambda}}) = -\mathcal{I}_{n,n_t,\hat{\lambda}}
\tag{35}
$$

$$
(C) = E(\hat{g}_{n_t,\hat{\lambda}}) - E(\widetilde{g}_{n_t,\hat{\lambda}}) \xrightarrow[\delta']{\leq 2B\sqrt{\log(1/\delta')/2n_t}} E_{n_t}(\hat{g}_{n_t,\hat{\lambda}}) - E_{n_t}(\widetilde{g}_{n_t,\hat{\lambda}}) \leq 0
\tag{36}
$$

This gives us the following with failure probability $5\delta'$:

$$
E(\widetilde{f}_{n,\hat{\lambda}}) - E(\widetilde{g}_{n_t,\hat{\lambda}}) \leq \min\{D, E\}
\tag{37}
$$

$$
D = -\widetilde{\mathcal{I}}_{n,n_t,\hat{\lambda}} + 2B\sqrt{\frac{\log(1/\delta')}{2n}}
\tag{38}
$$

$$
E = \rho_{\mathsf{out}} - \mathcal{I}_{n,n_t,\hat{\lambda}} + 4B\sqrt{\frac{\log(1/\delta')}{2n}} + 2B\sqrt{\frac{\log(1/\delta')}{2n_t}}
\tag{39}
$$

With $B' = 2B\sqrt{\log(1/\delta')}$, we can rewrite above as

$$E(\widetilde{f}_{n,\hat\lambda}) - E(\widetilde{g}_{n_t,\hat\lambda}) \leq \min\left\{ -\widetilde{\mathcal{I}}_{n,n_t,\hat\lambda} + \frac{B'}{\sqrt{2n}}, \quad \rho_{\mathsf{out}} - \mathcal{I}_{n,n_t,\hat\lambda} + \frac{2B'}{\sqrt{2n}} + \frac{B'}{\sqrt{2n_t}} \right\} \quad (40)$$

$$\leq \frac{B'}{\sqrt{2n}} - \max\left\{ \widetilde{\mathcal{I}}_{n,n_t,\hat\lambda}, \quad \mathcal{I}_{n,n_t,\hat\lambda} - \rho_{\mathsf{out}} - \frac{B'}{\sqrt{2n}} - \frac{B'}{\sqrt{2n_t}} \right\} \quad (41)$$

with failure probability of $5\delta'$.

This combined with equations (29) & (30) using the union bound with $\delta' = \delta/(7+3|\Lambda|) = \delta/L_4$ gives us (31) with probability at least $1 - \delta$. ∎

### B.6 Details on Empirical Validation

**Synthetic data.** The binary classification data is generated using the `make_classification` function (Guyon, 2003) in the `datasets` module of `scikit-learn` (Pedregosa et al., 2011). We ensure that the classes are not overlapping and there is no label noise, ensuring that the Bayes optimal risk $E(f^\star) = 0$.

**HPO search space.** We considered 36 configurations for a fully connected multi-layered perceptron, varying (i) the depth, (ii) the number of neurons in each layer, (iii) the initial learning rate for the SGD optimizer, (iv) the batch size for SGD. The implementation is in `PyTorch` (Paszke et al., 2019).

**Bound computation.** For any HP $\lambda$, (i) the approximation error $\mathcal{E}_{\mathsf{app}}(\lambda)$ is estimated as the best test-set error during ERM on $\mathcal{F}_\lambda$ with training sets of different sizes and different ERM restarts, (ii) the maximum error discrepancy $\sup_{f\in\mathcal{F}_\lambda}|E_n(f) - E(f)| \leq \Delta(\mathcal{F}_\lambda, n, \delta)$ is approximated with the empirical Radamacher complexity for $\mathcal{F}_\lambda$. Thus, $\Delta(\mathcal{F}_\lambda, n, \delta)$ is defined by the following quantity:

$$\Delta(\mathcal{F}_\lambda, n, \delta) \doteq \max_{f\in\mathcal{F}_\lambda} \sum_{i=1}^{n} \sigma_i f(x_i) + \sqrt{\log(2/\delta)/n}, \quad \sigma_i = \begin{cases} +1 & \text{w. p. } 1/2, \\ -1 & \text{otherwise.} \end{cases} \quad (42)$$

#### B.6.1 Complete results

The complete results for all values of $\frac{n_v}{n} \in [0.1, 0.5]$ for Figure 1 and Figure 2 in Figure 3 and Figure 4 respectively.

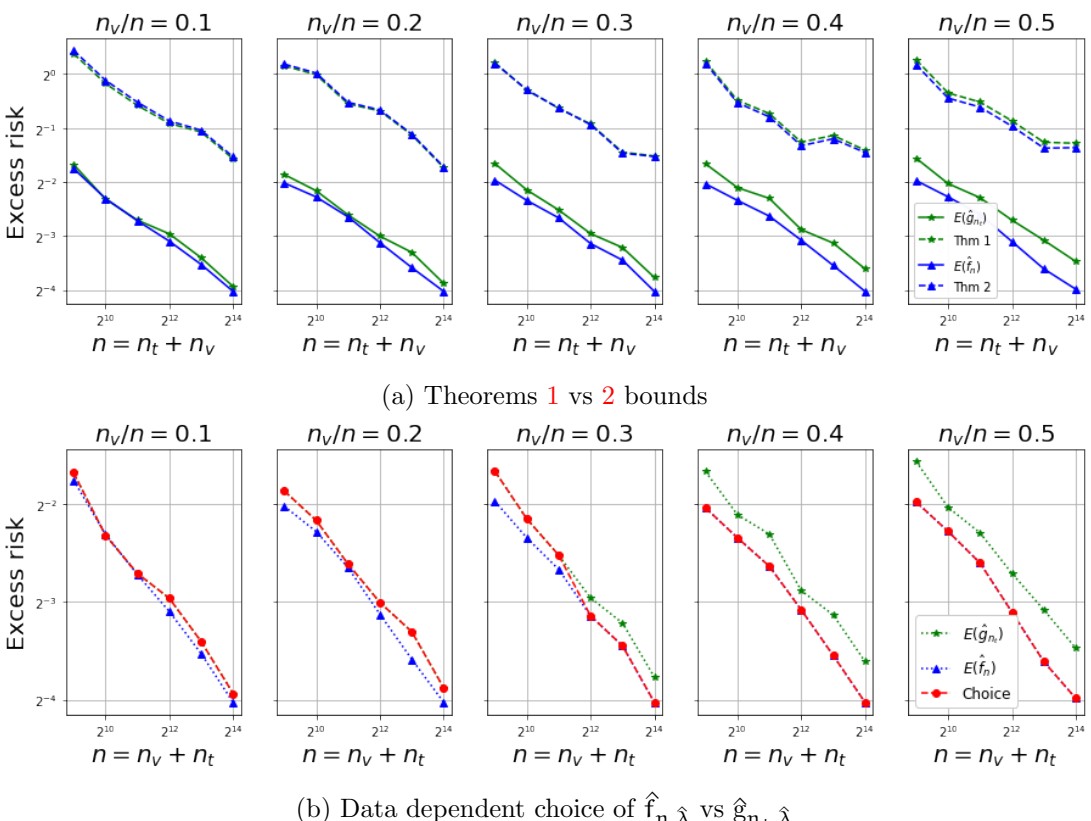

(a) Theorems 1 vs 2 bounds

(b) Data dependent choice of $\hat{\hat{f}}_{n,\hat{\lambda}}$ vs $\hat{\hat{g}}_{n_t,\hat{\lambda}}$

Figure 3: Empirical validation of presented theoretical results and proposed heuristic on synthetic data for a HPO problem for Neural Network configurations. Complete results for Figure 1.

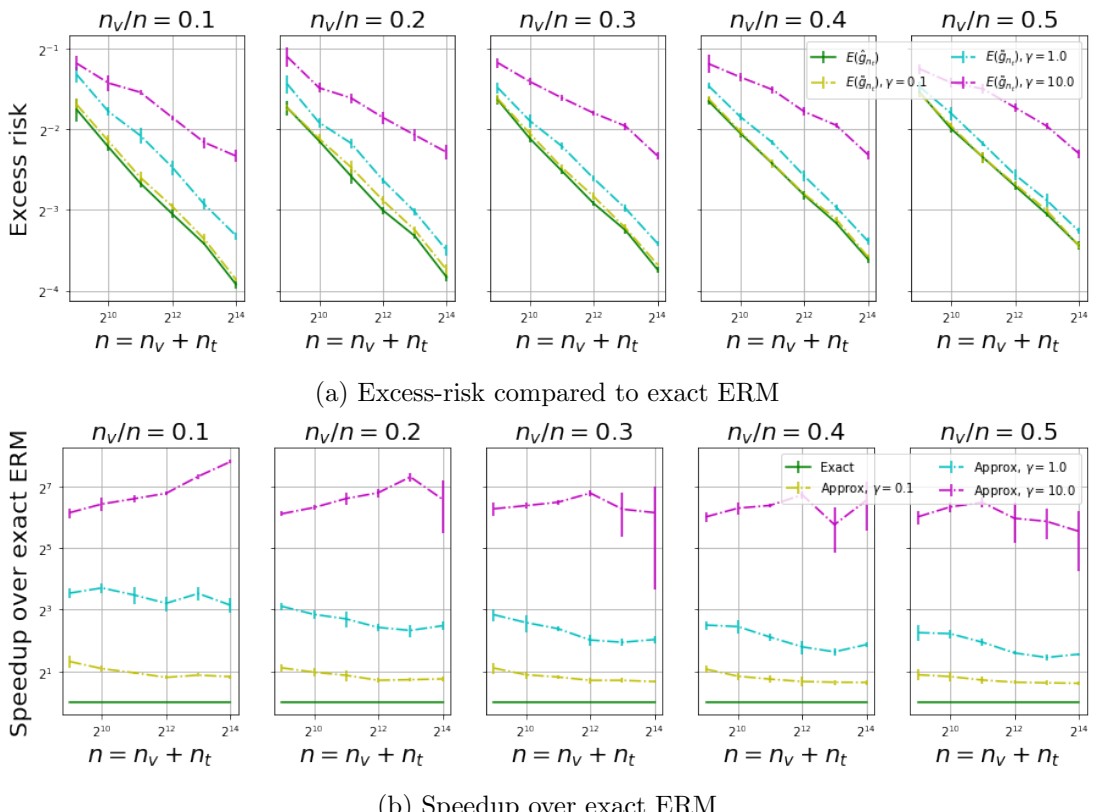

(a) Excess-risk compared to exact ERM

(b) Speedup over exact ERM

Figure 4: Empirical validation of the utility for data-dependent choice of $\rho_{\texttt{in}}$. Complete version of Figure 2.

