# OpenReview forum: "Leveraging Theoretical Tradeoffs in Hyperparameter Selection for Improved Empirical Performance"
_ICML.cc/2021/Workshop/AutoML — AutoML@ICML2021 Poster_

### Official Review · Reviewer_77y3 · 2021-06-15
**Review: Leveraging Theoretical Tradeoffs in Hyperparameter Selection for Improved Empirical Performance**

**Rating:** 7
**Confidence:** 1

**Review:**

The authors present new bounds for the risk incurred during hyperparameter optimization (including the situation where empirical risk minimization problems are solved only approximately).  Using these bounds, new heuristics are presented to help estimate the associated excess risk of a model fitting.

This is a pretty cool paper.  I think that it probably belongs somewhere a bit more substantial than just a workshop, and I hope that others can provide more insights as to the possible directions of improvement (for publication in a larger forum).  As for myself, I can only say that I found the presentation very impressive, and that I lack the necessary amount of time required to actually dive into these proofs in the context of a workshop review.  The graphs were a nice touch -- unexpected (and quite pleasing) given that the article is an entirely theoretical article.

---

### Official Review · Reviewer_MoXH · 2021-06-17
**Towards theory for HPO with practical relevance**

**Rating:** 7
**Confidence:** 2

**Review:**

**summary:** The paper presents theoretical bounds on "excess risk" in settings that resemble real-world practical HPO scenarios,  heuristics exploiting these bounds, and provides some empirical evidence of the utility thereof.

**assessment:** Theory for HPO is limited, and theory that is actually relevant in practice even rarer, and it is my belief that HPO research and practice would benefit from this line of work. That being said, I sadly did not have the time/expertise to confidently assess this work's impact or rigor. However, the same probably holds for the majority of HPO researchers/practitioners, and I feel that some core concepts (e.g., excess risk) could have been introduced more gently to make the work more accessible and attract wider attention. Also, the paper would be stronger if it considered an even more realistic HPO setting (> 'HP selection', e.g. BO/multi-fidelity, etc.) and would include empirical comparisons with state-of-the-art HPO methods and extensions thereof.

---

### Official Review · Reviewer_7cTk · 2021-06-18
**Interesting approach to a meaningful problem but experiments can be fleshed out more**

**Rating:** 6
**Confidence:** 4

**Review:**

This paper provides theoretical guidance for how long to train a model on a large dataset after performing hyperparameter optimization on a subset of the data.

Pros:
- Clearly written.
- Interesting approach to a meaningful problem.

Cons:
- Limited empirical results.  Results in the paper are limited to a synthetic generated dataset in the small and large-data settings.  It is unclear how well this approach will work on real world problems.
- Difficult to apply in practice.  Requires tuning a scale parameter to determine how far to train the inner loop to identify \hat{\lambda}.  Additionally,  it is unclear how the proposed approach will perform for large search spaces with many hyperparameter settings.

---

### Decision · Program_Chairs · 2021-06-21

Accept (Poster)